# Green Extraction by Ultrasound, Microencapsulation by Spray Drying and Antioxidant Activity of the Tucuma Coproduct (*Astrocaryum vulgare* Mart.) Almonds

**DOI:** 10.3390/biom11040545

**Published:** 2021-04-08

**Authors:** Lindalva Maria de Meneses Costa Ferreira, Rayanne Rocha Pereira, Fernanda Brito de Carvalho, Alberdan Silva Santos, Roseane Maria Ribeiro-Costa, José Otávio Carréra Silva Júnior

**Affiliations:** 1Laboratory of Pharmaceutical and Cosmetic R&D, College of Pharmacy, Federal University of Pará, Belém 66075-110, Brazil; lyndalva47@hotmail.com (L.M.d.M.C.F.); rayannerocha@yahoo.com (R.R.P.); 2Laboratory of Innovation and Development of Pharmaceutical Technology, Federal University of Amazonas, Manaus 69067-005, Brazil; fernandabc_@hotmail.com; 3Laboratory of Systematic Investigation in Biotechnology and Molecular Biodiversity, Federal University of Pará, Belém 66075-110, Brazil; alberdan@ufpa.br; 4Laboratory of Pharmaceutical Nanotechnology, Federal University of Pará, Belém 66075-110, Brazil; rmrc@ufpa.br

**Keywords:** ultrasound extraction, agroindustrial coproduct, tucuma, microencapsulation, spray drying

## Abstract

The industrial processing amazon fruits, like tucuma, generates a large amount of coproducts with large nutritional potential. Thus, this work obtained the oily extract of the tucuma almonds coproducts by green extraction using palm oil by the ultrasound method and then microencapsulated by atomization and verification of its antioxidant activity. Thermogravimetric techniques, infrared spectroscopy, scanning electron microscopy, moisture content, water activity were applied to characterize the microparticles. Total carotenoids were determined by UV spectroscopy and antioxidant activity was measured by 2,2′-azino-di-(3-ethylbenzthiazoline sulfonic acid and co-oxidation in the system β-carotene/linoleic acid. The oily extract and microparticle had total carotenoid contents of 3.305 mg/100 g ± 0.01 and 2.559 mg/100 g ± 0.01, respectively. The antioxidant activity assessed through the 2,2′-azino-di-(3-ethylbenzthiazoline sulfonic acid value was 584.75 μM/trolox ± 0.01 (oily extract) and 537.12 μM/trolox ± 0.01 (microparticle) were determined. In the system β-carotene/linoleic acid showed oxidation of 49.9% ± 1.8 lipophilic extract and 43.3% ± 2.3 microparticle. The results showed that the oily extract of the tucuma almond coproduct can be used as a carotenoid-rich source and microencapsuled with possible application for functional foods production.

## 1. Introduction

The tucuma (*Astrocaryum vulgare* Mart.) is a palm tree belonging to the Arecaceae family and has as its fruit a drupe with the mesocarp and epicarp of color ranging from yellow to dark orange related to the presence of carotenoids [1]. The food industry uses the fruit pulp in the preparation of creams and ice creams; the oil industry uses the almond and the pulp of tucuma for extracted fat and vegetable oils, these processes generate a large amount of coproducts that are discarded in the environment without any prior treatment. However, there are several studies in the literature [2,3] in which residues or by-products of plant origin are used as substrates because they still contain active principles. In this sense, the residues or by-product that would be discarded become util and can be reusing in the production chain [4,5,6,7].

The coproduct tucuma and fruit are very high sources of carotenoids (as β-carotene, all- trans- β-carotene, 13-cis- β-carotene, all-trans-α-carotene, and all-trans- β-cryptoxanthin). From this perspective, they can be widely used by the pharmacutical, cosmetics and food industries because they contain secondary metabolites with high levels of antioxidants. [8,9,10]. Carotenoids have several functions in human nutrition and health, but humans cannot synthesize them, thus they should be obtained from the diet or via supplementation. Among so many benefits to human health, they are extremely important for their antioxidant capacity and for being a precursor of vitamin A [11].

The carotenoids are naturally lipophilic, so they are easily extracted by organic solvents such as acetone, alcohol, chloroform, ethyl ether, petroleum ether and hexane. However, these organic solvents are toxic both to humans and animals, so, the use of these solvents is not well accepted in the food, pharmaceutical and cosmetic industries [12,13]. The green extraction can be a good alternative. The green extraction is based on experimentation and design of extraction processes that reduce energy consumption, allow the use of alternative solvents and renewable/innovative vegetable sources for example vegetable oils (palm, canola, soy, sunflower, among others) in order to eliminate toxic solvents and ensure safety and high quality of the extracted product [13,14,15,16].

This manner, the use of vegetable oils is a promising alternative in the extraction of carotenoids, due to the low polarity and solubility of these substances allowing them to be attracted by the vegetable oil during the extraction process [13,15,17,18,19]. The use of these oils as a solvent has demonstrated benefits, because they reduce the risks to health because they are not toxic, preserve the organoleptic characteristics of the extracted product, do not harm the environment and so can be viewed as a beneficial alternative in a sustainable world [13,20]. Moreover, the oil performs a barrier role against oxygen and consequently delays the oxidation time and the degradation rate of carotenoid extract [16].

Razi et al. [15] performed the extraction of astaxanthin from shrimp processing residues using sunflower oil and sunflower oil methyl ester as two green solvents. Sachindra and Mehendrakar [18] studied the capacity of extracting carotenoids from shrimp residues in different vegetable oils (sunflower, peanut, ginger, mustard, soy, coconut and rice flour) and optimized conditions such as the proportion of oil to residues, heating time and temperature. Handayani et al. [21] investigated the extraction of the astaxanthin carotenoid present in shrimp residues, using palm oil as an efficient solvent. Li et al. [13] investigated the extraction of β-carotene from the residues of pomegranate peel with sunflower seed and soybean oils and concluded the efficiency of the process.

However, despite several studies that show the efficiency of the use of vegetable oils in extractive processes, the disadvantage is the high viscosity that results in low diffusivity and, consequently, low extraction performance even at high temperatures. In this perspective, in recent years, ultrasound-assisted extraction of compounds with antioxidant activities has been widely applied to alleviate this problem [13,22]. Ultrasound extraction is considered a clean technology in the food industry because it is an effective method for the extraction of chemical constituents from plant matrices [23], low cost and easy operation [24,25]. The increase in ultrasound extraction is attributed to the propagation of ultrasound pressure waves and resulting cavitation forces, where bubbles can collapse explosively and generate localized pressure, causing rupture of plant tissue and improving the release of intracellular substances in the solvent [26]. The use of ultrasound for vegetal extraction brings benefits such as increased mass transfer, better penetration of the solvent, less dependence on the solvent used, extraction at lower temperatures, faster extraction rates and higher product yields [27]. These characteristics make sonication promising for many extractions including those in large scale [28].

Li et al. [13] developed a method for ultrasonic assisted extraction of carotenoids from fresh carrots in which sunflower oil was applied as a substitute for organic solvents. Luengo et al. [29] investigated the influence of moderate pressure application on the ultrasonic extraction of carotenoids from dried tomato pulp. Eh and Teor [30] analyzed the lycopene yield conditions of tomatoes and minimized degradation and isomerization during the ultrasonic extraction process. Almahy et al. [31] performed the extraction of carotenoids as natural colorants of the carrot by the ultrasound method. Zhang and Zelong [32] optimized and compared lycopene extraction from tomatoes by microwave-assisted extraction and ultrasound-assisted extraction in order to verify their advantages and disadvantages. Kumcuoglu and Tavman [33] investigated the lycopene ultrasound extraction of tomato processing residue and compared the effects of factors such as temperature, time, ultrasonic intensity and liquid-solid ratio with conventional extraction with organic solvents. Xu and Pan [34] evaluated the efficiency, productivity and selectivity of all trans-licopene present in the red grapefruit for ultrasound-assisted extraction.

The sensitivity of carotenoids to light, acidity, pH and pro-antioxidant compounds makes it necessary to use alternatives that protect these compounds, minimize losses and conserve bioactive properties [35]. In this perspective, the process of microencapsulation by spray drying has been successfully used in the food area, aiming to protect substances that have sensitivity to light, oxygen and storage time, such as carotenoids and due to its ease of operation and good cost-benefit ratio. Moreover, this process can prevent interactions with other compounds, promoting a greater stability of the product and consequently, increasing the useful life of the product [36,37].

In this sense, there are several indispensable factors to obtain dry extracts with better physicochemical characteristics and increase the operation yield. Thus, it is necessary to optimize the drying parameters such as inlet and outlet temperature and feed flow rate, concentration and type of technological adjuvants, and dry residue contents of the fluid extract to be dried [38]. Thus, the objective of this study was to obtain the oily extract from the coproduct of the tucuma kernels using the palm oil by the ultrasound technique in order to perform a new extraction alternative from the principles of green chemistry and then microencapsulate it by spray drying using maltodextrin and Arabic gum as encapsulant agents. In the same way evaluate their physical-chemical characteristics, determine the content of total carotenoids and their antioxidant capacity.

## 2. Materials and Methods

### 2.1. Chemicals, Reagents and Encapsulating Agents

2,2-azinobis (3-ethylbenzothiazolin-6)-sulfonic acid) (ABTS), 6-Hydroxy-2,5,7,8-tetramethylchroman-2-carboxylic acid (Trolox), Linoleic acid, β-carotene, polyoxyethylene orbitan monopalmitate (Tween 40) were obtained from Sigma-Aldrich (St. Louis, MO, USA). Potassium bromide (KBr) (potassium bromide 99+% for spectroscopy IV/Shimadzu). Maltodextrin with dextrose equivalent (DE 10) was acquired Corn Products (São Paulo, Brazil) and Synth pure acacia gum (São Paulo, Brazil).

### 2.2. Sample Preparation

Tucuma (*Astrocaryum vulgare* Mart.) seeds were undergone a baking process at 65 °C for 45 min, after which they were pressed to remove the oil to be industrially exploited. The resulting coproduct, designated as tucuma seed coproduct provided by the Amazon Oil Industry (Ananindeua, Brazil). After receiving it, the coproduct was packed and stored in a freezer (−18 °C). The material was dried for seven days in an oven with circulation and renewal of Air (SL-102 SOLAB, Piracicaba, São Paulo, Brazil) at a temperature of 40 °C ± 2 °C. During the whole period that the material was submitted to the oven and followed until the constant weight was obtained, in order to then, detterminate the end of the drying period. After dehydration, the dried material was weighed and ground in a knife mill and the powder of the dry coproduct of the tucuma almonds was obtained and stored in a freezer (−18 °C) until the moment of use.

### 2.3. Extraction of Carotenoids from the Coproduct

To obtain the oily extract for the extraction of carotenoids, the coproduct was placed in contact with the palm oil ceded by the Laboratory of products of animal origin (LAPOA) of the Federal University of Pará and submitted to ultrasound, in a Unique ultrasonic device (USC-2800, São Paulo, Brazil). The frequency 40 KHz and power US:154W with controlled temperature of 35 °C. The process was carried out by dispersing 10 g of the powder in 100 mL of refined palm oil in the proportion of 1:10 (*w*/*v*). The sample was then extracted in 30 min cycles for 4 h. The erlenmeyer was covered with aluminum foil to avoid exposure of the extract to ambient light [39].

### 2.4. Microparticle Preparation (MP)

The microparticles were obtained in two steps. The maltodextrin (MD) and Arabic gum (GA) were the encapsulating agents. The first step was the homogenization of the aqueous phase with the oily extract. The aqueous phase was composed by MD (5%) and GA (10%) both solubilized in distilled water at the final volume (200 mL), which was then added to the oily extract (5%) under magnetic stirring for 30 min [40]. Then, it was dispersed to the ultra turrax (IKA, T 125, São Paulo, Brazil) at 5000 rpm for 20 min for complete homogenization. The next step was the atomization of the emulsion obtained using a spray dryer (LM-MSDi 1.0 Labmaq do Brasil–LTDA, São Paulo, Brazil), according to the parameters: temperature of 100 °C; flow rate of 7.5 mL/min; and pressure of 6 bar. During the whole process the emulsion was kept under agitation to ensure homogeneity.

### 2.5. Encapsulation Efficiency (EE)

The EE of the carotenoids was calculated as the content of the carotenoids contained in the microparticle in relation to the content of carotenoids contained in the oily extract before drying [41]
%EE = (content of carotenoids in the microparticle)/(content of carotenoids in the extract before drying) × 100(1)

The total carotenoid content was extracted from the microparticles (0.4 g) with the addition of 2 mL of a methanol/water solution (50:50 *v*/*v*). The vortex dispersion was homogenized for 1 min and placed in ultrasonic bath (Cleaner Kondentech, São Paulo, Brazil) for 20 min. It was centrifuged for 15 min at 7500 rpm. The supernatant was filtered in membranes with a pore diameter of 0.45 µm (Millipore, Bedford, MA, USA) [42] for subsequent determination of the carotenoid content in a 470 nm UV-VIS (Shimadzu^®^ UV 1800, Kyoto, Japan) [43].

### 2.6. Quantification of Total Carotenoids by Spectrophotometry

The determination of the total carotenoids in the microparticles was performed by the UV/Vis spectrophotometric method (Shimadzu UV 1800, Kyoto, Japan). 3 g of the microparticle powder were weighed and dissolved in hexane to a final volume of 10 mL. The reading was performed at a wavelength of 470 nm using the specific coefficient for β-carotene (Eo = 2592). The analysis was also performed on the extract before drying. The results were performed in triplicate and expressed in mg carotenoids/100 g [43]. The results were calculated using the Equation (2):C = (Absorbance × solution volume × 10^6^)/(100 × 2592 × sample weight)(2)

### 2.7. ABTS Assay

The antioxidant activity by the free radical capture ABTS was performed in a spectrophotometer (Shimadzu^®^ UV 1800, Kyoto, Japan). 30 μL the solution obtained from the extraction of carotenoid content from the microparticles were mixed with 3000 μL of ABTS solution and incubated in the dark for 6 min followed by measurement of absorbance at 734 nm. The antioxidant activity was calculated based on a Trolox standard curve (100 μM–2000 μM) and the results were expressed in μM Trolox equivalent /g dry weight (TE) [44]. The analysis was also performed on the oily extract before drying in the same way that was performed on the microparticles.

### 2.8. β-Carotene/Linoleic Acid System Assay

The antioxidant activity in the β-carotene/Linoleic Acid System in a spectrophotometer (Shimadzu^®^ UV 1800, Kyoto, Japan). The β-carotene/Linoleic acid solution was prepared from a mixture of 40 μL of linoleic acid, 530 μL of Tween 40, and 50 μL of the β-carotene solution at a concentration of 2 mg/mL in chloroform and 500 μL in a conical flask. The mixture was subjected to complete evaporation of the chloroform under nitrogen. Then, the oxygen-treated water was added until an absorbance between 0.6 nm and 0.7 nm was obtained at a wavelength of 470 nm. The system solution is yellow-orange in color and must always be protected from light and readily used. Three different dilutions of the extract and microparticle were prepared in test tubes in triplicate. 0.4 mL of each dilution of the extract and microparticles was mixed with 5 mL of the system solution. 0.4 mL of the Trolox solution at a concentration of 200 mg/mL was used as a control with 5 mL of the β-carotene/linoleic acid system solution; the test tubes were homogenized, shaken, and kept in a water bath at 40 °C. The first reading was performed at 470 nm after 2 min of mixing and then at intervals of fifteen min to 120 min. The spectrophotometer was calibrated with water. The results were expressed as a percentage of oxidation inhibition. The reduction in the absorbance of the system without an antioxidant was considered as 100% oxidation. The decrease in the absorbance reading of the samples is correlated with the system and establishes the oxidation percentage, subtracting the oxidation percentage of each sample from 100 [45]. The analysis was also performed on the oily extract before drying in the same way that it was performed on the microparticle.

### 2.9. Physicochemical Characterization of the Microparticle

#### 2.9.1. Infrared Analysis (FTIR)

The FTIR spectrum was obtained by spectrophotometer (IR Prestige-21 Shimadzu^®^ Kyoto, Japan) performed by mixing the powder sample with potassium bromide (KBr) and pressing it at high pressure, forming a tablet. The absorption spectrum was analyzed in the range of 4000 to 600 cm^−1^, with 32 scans and a resolution of 4 cm^−1^ [40]. The samples submitted to FTIR analysis were the oily extract, the binary mixture (maltodextrin and Arabic gum and oily extract), and the microparticles. The binary mixture was prepared as follows: the adjuvants maltodextrin and Arabic gum (BM) and oily extract in the proportion of 1:2:1 (*w*/*w*) in gral and pistil. The spectrum was obtained using the Origin Pro2019 software.

#### 2.9.2. Thermogravimetry Analysis (TG)

The TG curves of the oily extract, binary mixture, and microparticles were obtained in a TGA-50 thermal analyzer (Shimadzu^®^, Kyoto, Japan) using a platinum crucible with approximately 9.0 mg of sample, under a nitrogen atmosphere (N_2_) and flow 50 mL/min. The experiment was carried out from room temperature to 600 °C and a heating rate of 10 °C/min. The data obtained were analyzed using the TA-50W Shimadzu^®^ software [46].

#### 2.9.3. Differential Scanning Calorimetry (DSC)

The DSC curves of the oily extract, binary mixture and microparticles was performed on DSC-60 plus equipment (Shimadzu^®^, Kyoto, Japan). Approximately 3 mg of the samples were deposited in an aluminum crucible. The analyzes were performed in a nitrogen atmosphere (50 mL/min) with and a heating rate of 10 °C/min and temperature of 300 °C [46]. The analysis was also performed on the oily extract and in the binary mixture.2.9.4. Moisture content.

The determination of the moisture content was evaluated by the gravimetric method on a scale for moisture analysis with a halogen lamp (GEHAKA, São Paulo, Brazil). Approximately 2 g of the microparticle powder was placed on a scale for moisture analysis with a halogen lamp at a temperature of 105 °C for a time of 15 min (in triplicate). The balance determined the exact value of the percentage of moisture loss [47].

#### 2.9.4. Water Activity

The water activity was determined using electronic equipment (Aqua Lab Dew Point 4TEV, São Paulo, Brazil) at a temperature of 25.03 °C. The operation was performed in triplicate [48,49].

#### 2.9.5. Particle Morphology and Particle Size

The morphological analysis and particle size (evaluated in triplicate) of the microparticle were performed using the Scanning Electron Microscope (Zeiss, EVO-MA-10, Jena, Germany). The microparticles were deposited on a sample holder with the aid of carbon adhesive tape and coated with a layer of gold (Au) 15 nm thick for 1.5 min and observed in secondary electrons and magnification 1000×, 5000×, 10,000×, and 12,000× [50].

### 2.10. Statistical Analysis

The results obtained from EE, antioxidant activity, total carotenoids, water activity, moisture content were analyzed and expressed as mean ± standard deviation, using Office 365 Excel. Analysis of variance (ANOVA) was used to compare the samples of carotenoid content and antioxidant activity and the differences between means were detected posteriorly by the Tukey test considering (*p* < 0.01) in the BioEstat 5.0 program (Brazil, Tefé, AM). All samples were analyzed in triplicate.

## 3. Results and Discussion

### 3.1. Quantification of Total Carotenoids, Encapsulation Efficiency and Antioxidant Activity

Carotenoids are nutrients generally reported in yellow to red fruits and vegetables, such as papaya, acerola, pupunha, carrot, and tucuma [8,51]. They are related to important physiological functions and actions, with pro-vitamin A being the best known. Besides, a positive correlation was observed between the intake of vegetables and fruits containing carotenoids and the prevention of several chronic-degenerative diseases, such as cancer, inflammation, cardiovascular diseases, cataracts, and age-related macular degeneration, among others [51]. Prudent levels of daily carotenoid intake recommended by Institute of Medicine [52] are 3000 to 6000 μg for β-carotene, 5200 to 6000 μg for pro-vitamin A carotenoids, and 9000 to 18,000 μg for total carotenoids [52,53].

Table 1 shows the total carotenoids value for oily extract and microparticle. The differences in total carotenoids content of the samples are statistically significant different (*p* < 0.01), oily extract tucuma coproduct (3.305 mg/g) showed a total of 0.746 mg more than the total carotenoids contained in the microparticles (2.559 mg/g) (Table 1). This difference is may be associated to a small loss by carotenoid degradation during the drying process [54]. In view of the encapsulation efficiency have been around 78%. For the first time, the carotenoid content was extraction by ultrasound of the co-product of the tucuma almonds (*Astrocaryum vulgare* Mart.) using a vegetable oil as an extractor liquid. Therefore, there are no data in the literature to be compared with the results found in this study. In this sense, the importance of the data obtained is emphasized. Considering that in this work we used the waste of the tucuma, we can infer that the carotenoid content found in our samples is promising.

Spray drying has been highlighted when it comes to procedures involving the microencapsulation of bioactive compounds because of their ease of operation and cost-effectiveness. By removing water from products, ensures microbiological stability, pH, prevents degradation and oxidation reactions, protects from moisture, reduces the cost of storage and transport, in addition to obtaining a product with specific properties, such as instant solubility [55,56]. After performing the spray dryer microencapsulation process, parameters such as microencapsulation efficiency are important to be evaluated to verify the efficiency of the process. There is an optimum concentration of the material to be encapsulated as way to guarantee greater yield [57].

Table 1 shows an encapsulation efficiency value of the microencapsulated extract of 77.42%, according to the EE analysis, the value shows that the carotenoids remained in more than 70% in the microparticles after the drying process. This means, that of the total of 2.559 mg/g total carotenoids, 1.981 mg/g are inside the microparticle or adsorbed on its wall. This value is higher than those reported by Álvarez-Henao [58] who obtained EE values for lutein microcapsules between 9.90% and 32.9% and astaxanthin, which were between 33.4% and 70.10% [59]. Arabic gum, because of its structural characteristics, has an amphiphilic character, which allows its absorption on lipophilic surfaces, acting as a protective colloid and, therefore, a good film-forming agent. In addition, it presents low viscosity and Newtonian behavior in concentrations below 35% and thus is one of the most effective film-forming materials for encapsulation [60]. Due to the encapsulation efficiency, Arabic gum has been normally used to encapsulate lipids [61].

Carotenoids are one of nature’s main antioxidant pigments, found in many fruits. Reactive oxygen species (ROS) are highly reactive molecules and the body controls its degradation through two integrated antioxidant systems: an endogenous enzyme system and exogenous control by the entry of non-enzymatic antioxidant molecules derived from the diet or produced by the body [62]. Thereby, the antioxidant properties of carotenoids are based on the structure of these compounds, mainly in the system of double conjugated bonds, also responsible for the color of these pigments, making possible the sequestration of free radicals, the modulation of the carcinoma metabolism, inhibition of cell proliferation, increased cell differentiation via retinoids, stimulation of communication between cells, immune strengthening, reduced risk of cataracts and skin protection caused by damage from UV radiation [63].

From this perspective, the evaluation of antioxidant activity by ABTS and β-carotene/linoleic acid in oily extract (584.75 μM/Trolox; 49.9%) and microparticles (537.12 μM/Trolox; 43.3%), respectively (Table 1) demonstrated that both showed good activity. The data obtained for both methods showed a statistically significant difference (*p* < 0.01).

Thus, the results showed that the carotenoid content after drying was preserved in the microparticles, which was confirmed in the EE. It is worth mentioning that the presence of oil has a great influence on the formation of particles, on the encapsulation efficiency, and provides protection to bioactive compounds. As well, the antioxidant activity by the two tested methods also corroborated to certify that the properties of interest were preserved.

### 3.2. Infrared Analysis

Spectroscopy is a technique that aims to identify (or even determine) functional groups (characteristic) of the organic compounds analyzed, providing a preliminary study of their chemical structure of isolated substances and complex samples [64]. It has been a technique used to provide important information on drug-excipient compatibility and possible reactions among the chemical species involved during or after any physical process [65].

The FTIR spectra of the oily extract, binary mixture, and microparticle are shown in Figure 1. The spectra obtained from the oily extract and binary mixture showed bands in 2909 cm^−1^ and 2918 cm^−1^ for the CH stretch. Bands at 1743 cm^−1^ and 1734 cm^−1^ related to vibrations of the C=O stretch, at 1447 cm^−1^ and 1444 cm^−1^ referring to the CH_3_ group folding vibration and at 1140 cm^−1^ and 1162 cm^−1^ are characteristic of CO stretching vibrations (Figure 1). However, the spectrum obtained for microparticles showed a band at 3047 cm^−1^ characteristic of OH stretch vibration and 2918 cm^−1^ referring CH, at 1743 cm^−1^ related to C=O stretch vibrations, at 1606 cm^−1^ related to C=C stretch vibrations and in 1034 cm^−1^ referring to C-C stretching vibrations [40]. The binary mixture was produced in the same proportions of adjuvants maltodextrin and Arabic gum and oily extract (1:2:1) used in the formation of the microparticle.

The spectra obtained in this work, presented an aggregate of absorption bands related to stretches C-H, C=O, C-O, C=C, CH_3_, OH, C-C that may be related to important functional groups such as alcohols, phenols, alkanes, alkenes, methyl, aromatic compounds and anhydroglucose ring, corresponding to maltodextrin [5]. The analysis of characteristic bands of certain functional groups of a molecule provides, through a simple examination of the spectrum and consultation of tables and data a set of information on the structure of the molecule [64]. Thus, it was possible through the FTIR spectra to obtain structural information of the molecules and detect significant changes in the shape and position of the absorption bands of the different functional groups of the microparticle or free molecules [66]. Overall, in the spectra of the binary mixture and from the oily extract there is almost no difference. The oily extract, binary mixture presented similar bands in 2918 cm^−1^ and band between 1734 cm^−1^ and 1743 cm^−1^ but in the microparticle with less intensity which may be related to the encapsulation of the oily extract [67,68]. The spectra obtained showed that the changes occurred did not interfere in the profile of the extracts by ensuring the permanence of the bands attributed to them. In this sense, FTIR contributes as a structural characterization technique in order to confirm the presence of important functional groups oily extract and in the microparticles.

### 3.3. Thermogravimetry Analysis (TG/DTG)

Figure 2 shows the TG/DTG curves of the oily extract, binary mixture and microparticle. The oily extract showed only one mass loss event that occurred in the temperature range of 402.78 °C–453 °C with 97.825% of mass loss (Figure 2). The thermal profile of the particles showed three mass loss events, the first occurred in the range of 48.23 °C −83.92 °C with 4.5% loss of mass; the second took place in the range of 251.54 °C–286.76 °C with 14.29% loss of mass; and the third event occurred in the temperature range of 329.94 °C–309.01 °C, with a loss of mass of 59.080% (Figure 2). The binary mixture revealed only two events: the first event with a loss of mass of 10,357% that occurred in the range of 38.78 °C–78.74 °C and the second occurred in the range of 299.86 °C–351.56 °C and loss of mass of 64.595% (Figure 2).

Among the three samples, the oily extract showed the lowest percentage of residue, only 2.175%, and the greatest loss of mass, which indicates its lower thermal stability. Meanwhile, the microparticle and binary mixture showed a residue percentage of 23.13% and 24.048% the bigger quantity of the residue for this samples probably is because Arabic gum and maltodextrine were not present in oil extract.

There was an event before 100 °C for the microparticle (48.23 °C) and binary mixture (38.78 °C) that is probably due to the release of water and/or volatile substances [40]. This difference of 10 °C may be related to the barrier against the release of assets that the wall material offers to the microparticle, delaying the exit of components of the oil extract of the tucuma. Since the mixture was lost in this first event, 10.357% of the mass, while the microparticle lost 4.5% of the mass and thus corroborates the hypothesis that the particle can protect the oil extract.

### 3.4. Differential Scanning Calorimetry (DSC)

The DSC curves of the oil extract, binary mixture and microparticle are shown in Figure 3. The DSC curve of the oil extract occurred in only one endothermic event in the range of 38–45 °C, with ΔH = 111.64 J/g (Figure 3). The DSC curve of the microparticle showed three endothermic events: the first in the range of 29–102 °C, with ΔH = 113.68 J/g and 65.47 °C Tg (Figure 3); second in the range of 200.75–237.04 °C, with ΔH = 8.70 J/g, with the peak temperature at 220.39 °C related to polymer melting; and the third event in the 243–282 °C range, with ΔH = 13.78 J/g, related to material degradation (Figure 3).

However, for the binary mixture, two events were recorded, an endothermic event in the range of 26–80 °C with ΔH = 114.22 J/g, and Tg of 53.70 °C and the other in the range of 210–273 °C, with ΔH = 103.36 J/g peak temperature at 210.75 °C related to the melting point of the polymers (Figure 3). The first event that occurred for the three samples is related to water evaporation. The energy expenditure that occurred during the water evaporation process in the microparticle and binary mixture was higher when compared to the oily extract. This increase in energy expenditure, probably, may be related to the connection of water with the adjuvants forming a matrix and, thus, hindering the water evaporation process. Indeed, the wall material, in addition to protecting sensitive food compounds during drying, can promote an increase in Tg and reduce the hygroscopicity of powders [69]. The Tg value must be greater than 40 °C to obtain long-term stability [70]. The microparticle Tg showed a value of 65.47 °C, that is, the adjuvants fulfilled their role of protecting the sensitive compounds and guaranteeing the stability of the product and, thus, showing the efficiency of the encapsulation process.

### 3.5. Moisture Content and Water Activity

The moisture content and water activity influence the efficiency of the drying process and affect the quality of the dry powder and the shelf life of food products [67,71]. In food products, the water activity value close to 0.3 indicates stability against non-enzymatic browning, microorganism development, and enzymatic activity during adequate food storage [72]. The moisture content can also change in particle size and morphological differences [73]. Foods that have a moisture content of more than 20% and a higher water activity of 0.60 are subject to deterioration processes caused by molds and yeasts [74]. The values found for the moisture content of 6.6% ± 0.06 and water activity 0.25 ± 0.007 were low and indicate a good drying process [75]. These values were similar to the study of microencapsulation of lutein (water activity that varied between 0.12 and 0.33 and moisture content varied between 4.21% and 9.01%) [58] and astaxanthin (water activity between 0.27 and 0.35 and moisture content varied) between 8.21% and 11.17%) [59].

### 3.6. Particle Morphology and Particle Size 

A morphological analysis was carried out to evaluate the characteristics of the microparticles such as shape, surface, as well as their size distribution (Figure 4). Microphotographs generally showed spherical particles of heterogeneous sizes and without agglomeration, which supposes that there is a repulsion due to negative charges. One can also see in Figure 4 that there are microparticles with a smooth surface and others that have a wrinkled surface.

The wrinkling can be attributed to the drying and cooling process of the particles in the spray dryer [76]. This roughness is probably influenced by the drying speed and the viscoelastic properties of the wall material [77]. The particles showed heterogeneity and were polydispersed and, therefore, suggests a greater variation in their properties, mainly in the solubility of the particles in a food matrix [78]. In this sense, the particle size is related to the amount of adjuvant, and formulations with a lower proportion of wall material showed a small decrease in the average diameter [61]. The value of the particle size found was 15.89 µm ± 33.47 considered adequate because it is a microparticle. This size make the product highly soluble, while, on the other hand, it also makes them more susceptible to oxidation.

## 4. Conclusions

The oily extract obtained from the tucuma almonds coproduct followed the principles of green chemistry without the need to use organic solvents. At the time, Palm oil was used as a new alternative for extracting lipophilic substances. The oily extract and the microparticle showed significant levels of carotenoids and good antioxidant activity by the tested methods. The microparticle showed spherical and heterogeneous structures, good encapsulation efficiency from the spray drying process using malt dextrin and gum Arabic as a wall material. The microparticle also had low humidity and water activity, an indication of good stability and conservation. Thus, it is suggested that the extraction and drying process were efficient and kept the antioxidant activity preserved, generating a product rich in carotenoids with possible application in the food area a functional food.

## Figures and Tables

**Figure 1 biomolecules-11-00545-f001:**
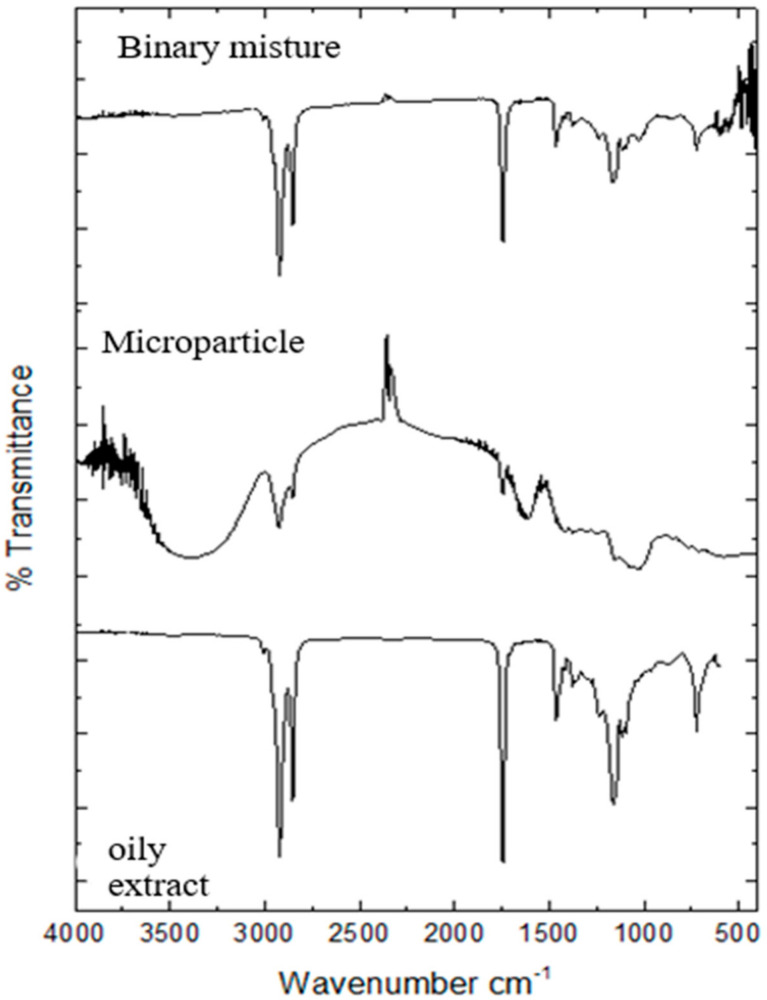
FTIR spectra of the binary mixture of adjuvants, microparticles and oil extract in the absorption range of 4000 to 600 cm^−1^, with 32 scans and 4 cm^−1^ resolution.

**Figure 2 biomolecules-11-00545-f002:**
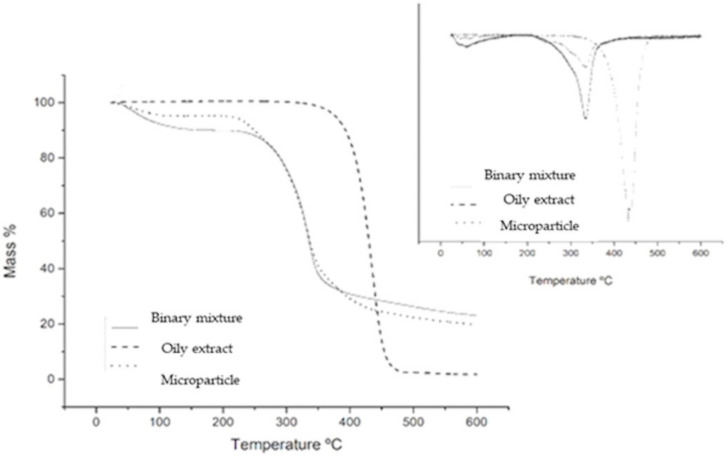
TG/DTG curve of binary mixture of adjuvants, the oily extract and microparticle. Conditions: nitrogen atmosphere (N_2_), the flow of 50 mL/min, and a heating rate of 10 °C/min.

**Figure 3 biomolecules-11-00545-f003:**
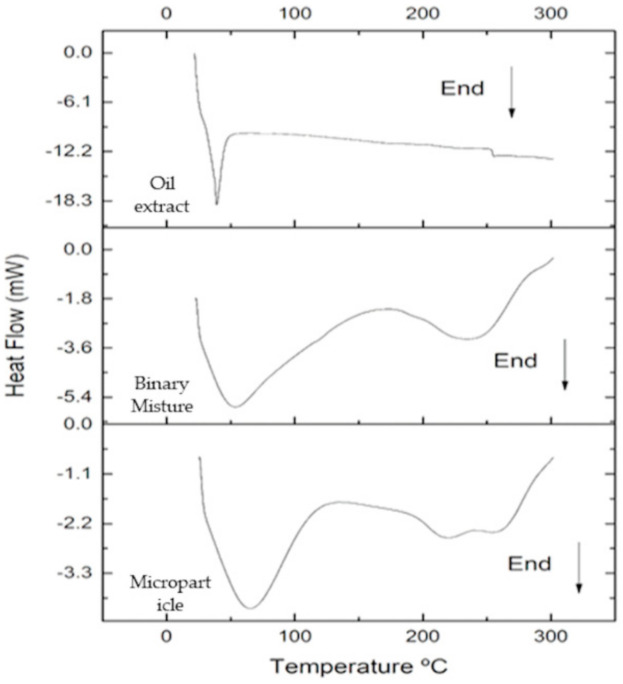
DSC curve of the oily extract, binary mixture of adjuvants and microparticle. Conditions: nitrogen atmosphere (N_2_), the flow of 50 mL/min, the heating rate of 10 °C/min and a temperature of 300 °C.

**Figure 4 biomolecules-11-00545-f004:**
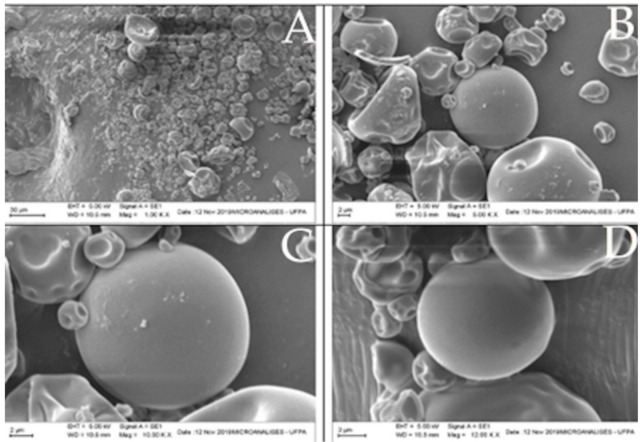
Microparticle photomicrographs: (**A**) (1000-× magnitude), (**B**) (5000-× magnitude), (**C**) (10,000-× magnitude) and (**D**) (12,000-× magnitude).

**Table 1 biomolecules-11-00545-t001:** Total carotenoids, encapsulation efficiency and antioxidant activity in the oil extract (OE) and microparticle (MP)**.**

Sample	Total Carotenoids (mg/g)	Encapsulation Efficiency (%)	ABTS^+^ (µM trolox)	β-Carotene/ Linoleic Acid (%)
OE	3.305 ± 0.01 ^a^		584.75 ± 0.00 ^a^	49.9 ± 1.8 ^a^
MP	2.559 ± 0.01 ^b^	77.42 ± 5.44	537.12 ± 0.01 ^b^	43.3 ± 2.3 ^b^

Values mean ± standard deviation (*n* = 3); Different letters in the same column indicate significant difference (*p* < 0.01) OE = oily extract; MP = microparticle.

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
