# Peer review of "Green Extraction by Ultrasound, Microencapsulation by Spray Drying and Antioxidant Activity of the Tucuma Coproduct (Astrocaryum vulgare Mart.) Almonds"

_biomolecules, 2021, doi:10.3390/biom11040545_

Round 1

Reviewer 1 Report

The oily extract obtained from the tucuma almonds coproduct have followed the principles of green chemistry without the need to use organic solvents and palm oil was used as an alternative for extracting lipophilic substances. The oily extract and the microparticle showed significant levels of carotenoids and good antioxidant activity by the tested methods. The microparticle showed good encapsulation efficiency from the spray drying process, good stability and conservation. Thus, the results suggested that the extraction and drying process were efficient and kept the antioxidant activity preserved, generating a product rich in carotenoids. The possible application in the food area a functional food is proposed.  The paper is well written, the tables and figures are clear and the conclusion are adjusted to the results obtained. The bibliography is updated. 

Only a minor comment:

line 138. The temperature -18º is very low to be considered refrigeration.

Author Response

Dear editors and reviewers,

We respectfully welcome and appreciate the valuable contributions of the reviewers, with their respective suggestions and requests for correction that certainly qualify and give greater consistency to our text.

Comments and Suggestions for Authors

The oily extract obtained from the tucuma almonds coproduct havefollowed the principles of green chemistry without the need to useorganic solvents and palm oil was used as an alternative forextracting lipophilic substances. The oily extract and themicroparticle showed significant levels of carotenoids and goodantioxidant activity by the tested methods. The microparticleshowed good encapsulation efficiency from the spray dryingprocess, good stability and conservation. Thus, the resultssuggested that the extraction and drying process were efficient andkept the antioxidant activity preserved, generating a product rich incarotenoids. The possible application in the food area a functionalfood is proposed. The paper is well written, the tables and figuresare clear and the conclusion are adjusted to the results obtained.The bibliography is updated.

We thank you for the time spent and for all the comments made to the manuscript.

Response to Reviewer 1 Comments

Only a minor comment:

Point 1: line 138. The temperature -18º is very low to be considered refrigeration.

Response 1: The authors agree with the reviewer. The sample was kept in freezing. Alteration was made in the manuscript.

Reviewer 2 Report

Dear Authors,

The study examines the possibility of extracting carotenoids from the coproduct of tucuma kernels using palm oil, and its microencapsulation by spray drying using maltodextrin and gum arabic as encapsulant agents.

My overall opinion is that the manuscript is very poorly designed and written.

I will list a few critical points that indicate very poor quality of work:

Line 41-42: “because that the material which the material that would be…” - bad writing style

Line 48: “ingest them in food” - bad writing style

Line 273-274: “This difference is probably associated to the atomization process, variables such as inlet temperature can be affect the content in the microparticle [53]” - Speculation, the exact reason for the difference in carotenoid content is not known because different microencapsulation conditions have not been investigated.

Line277-278: “carotenoid values in the pulp tucumã fruit (8.39 mg / 100g) and 5,46 times lower

the carotenoid values in the pulp tucumã fruit (18.06 mg / 100g) [8]” - Unclear information - the result of this study was twice compared with carotenoid values in the pulp of Tacuma fruit.

Line 280-281: “This is further supported when we compare the carotenoid content of our samples with the carotenoid content of pomegranate peel.” - Unclear and unnecessary comparison with the content of carotenoids from another source or raw material.

Line 338-339: “The binary mixture was produced in the same proportions of oily extract and adjuvants used in the formation of the microparticle.” - Unclear - isn't the composition of the binary mix just maltodextrin and gum arabic as stated in line 219-221.

Line 426-433: The Results and Discussion are incomplete - there is no data and discussion on the condition of the particles in which some other formulations of excipients or other drying conditions have been applied.

The work is very poorly designed, there is still a lot of analysis and testing to discuss and conclude about the optimal way of extraction, microencapsulation, and the possibilities of applying these methods. The recommendation is that the manuscript needs to be rejected.

Author Response

Dear editors and reviewers,

We respectfully welcome and appreciate the valuable contributions of the reviewers, with their respective suggestions and requests for correction that certainly qualify and give greater consistency to our text.

Comments and Suggestions for Authors

The study examines the possibility of extracting carotenoids from the coproduct of tucuma kernels using palm oil, and its microencapsulation by spray drying using maltodextrin and gum arabic as encapsulant agents.

My overall opinion is that the manuscript is very poorly designed and written.

We thank you for the time spent and for all the comments made to the manuscript.

Response to Reviewer 2 Comments

Point 1: Line 41-42: “because that the material which the material that would be…” - bad writing style

Response 1: The authors agree with the reviewer. The authors revised the text in the manuscript.

Point 2: Line 48: “ingest them in food” - bad writing style

Response 2: The authors agree with the reviewer. The authors revised the text in the manuscript.

Point 3: Line 273-274: “This difference is probably associated to the atomization process, variables such as inlet temperature can be affect the content in the microparticle [53]” - Speculation, the exact reason for the difference in carotenoid content is not known because different microencapsulation conditions have not been investigated.

Response 3: Response 3: The authors agree with the reviewer. There's been a mistake. No evaluation was performed under different conditions for the statement. The manuscript was amended.

Point 4:Line277-278: “carotenoid values in the pulp tucumã fruit (8.39 mg / 100g) and 5,46 times lower the carotenoid values in the pulp tucumã fruit (18.06 mg / 100g) [8]” - Unclear information - the result of this study was twice compared with carotenoid values in the pulp of Tacuma fruit.

Response 4: The authors compared with the fruit and pulp tucuma because there were no data in the literature related to the content of carotenoids extracted from the co-product of the tucumã almonds using palm oil as a liquid extractor. However, the authors decided to remove the comparison of the manuscript, as it is not a co-product, but other parts of the fruit, which have very different values.

Point 5: Line 280-281: “This is further supported when we compare the carotenoid content of our samples with the carotenoid content of pomegranate peel.” - Unclear and unnecessary comparison with the content of carotenoids from another source or raw material.

Response 5: There was a mistake in comparing the results of this study with different species. This was due to the lack of studies in the literature in which carotenoides were extraction of the co-product of tucuma almonds using vegetable oil as an ultrasound extractor liquid. The choice of study of pomegranate bark was due to the extraction of carotenoides using vegetable oil and by ultrasound. However, the authors removed the text the manuscritpt and emphasize the importance of the results obtained in this study.

Point 6: Line 338-339: “The binary mixture was produced in the same proportions of oily extract and adjuvants used in the formation of the microparticle.” - Unclear - isn't the composition of the binary mix just maltodextrin and gum arabic as stated in line 219-221.

Response 6: The binary mixture was made with maltodextrin and gum arabic (both encapsulating agents) as well as the oily extract, all in the same proportion in which the microparticle was prepared. There was an error in the writing of the sentence. The authors amended it in the manuscript.

Point 7: Line 426-433: The Results and Discussion are incomplete - there is no data and discussion on the condition of the particles in which some other formulations of excipients or other drying conditions have been applied.

Response 7: The authors agree with the reviewer. However the authors did not fully understand what the reviewer is suggesting. In the sense, they added to the results and discussion what they judged is incomplete.

Reviewer 3 Report

Abstract:

Pag 1, line 16: Please delete the comma after generates

Pag 1, line 17: Please replace “potencial” with “potential”

Pag 1, line 24: Rewrite as “The antioxidant activity assessed through the ABTS value was ….”

Pag 1, line 27-29: Rewrite as “The results showed that the oily extract of the tucumã almond coproduct can be used as a carotenoid-rich source and microencapsulated with possible application for functional foods production

Since most of the microparticle’s properties depend, inter alia, on the coating material selection please justify the choice

Throughout the text uniform the coproduct or co-product form

Introduction:

Pag 1, line 37: Please replace “use” with “uses”

Pag 1, line 41-43:  Please rewrite the following sentence: However, there are several studies in literature [2,3] in which residues or by- products of plant origin are used as substrates still containing active principles, because that the material which the material that would be thrown away become util and can be reusing in the production chain

Pag 2, line 44: Please close the round bracket

Pag 2, line 47: Seems that the choice of using the verb to be is not appropriate in the sentence Carotenoids are several functions. Please modify.

Pag 2, line 75: Add dash in beta -carotene

Pag 2, line 95: The reference’s style reported is not correct. Please delete the first letter of the authors' name

Pag 3, line 98-106: The references style reported in this part is not correct. Please delete the first letter of the authors' name

Pag 3, line 110: Replace “by drying spray” with “by spray drying”

Pag 3, line 122:  please uniform the form of Arabic gum in the text.

Materials and Methods

Pag 3, line 133: The title “Sample Obtaining” must be changed to better specify the paragraph content.

Pag 3, line 145: Was the Oily extract obtained after the almonds mechanical press as reported in the previous paragraph? The title of this paragraph is not appropriate. Please refer to the extraction of carotenoids from the co-product

Pag 3, line 139: throughout the text, make the temperatures uniform by adding a space between the number and the symbol (40 °C)

Pag 3, line 140-142: Please rewrite the sentence

Pag 4, line 158: the spelling of solubilized, please correct

Please reverse the paragraphs 2.5-2.6 order. In the paragraphs Encapsulation efficiency, after the explanation of the extraction methods, please report the quantification of the total carotenoids by spectrophotometry

In all the equations reported, please adjust the number. The same equation is marked with two numbers.

Pag 4, line 177: add a space between the number and the symbol (0.4 g)

Pag 4, line 186: is there 30 uL referred to the microparticle’s extract? Please correct

Pag 5, line 192: Please delete the full stop in β-carotene word

Pag 5, line 193-205: Please uniform the ml format in all the paragraph

Pag 5, line 214-222: Please add the software used to obtain the FT-IR spectrum

Pag 5, line 273: Is the amount express as mg/g (as reported in the table) or mg/100 g (as reported in the text.

Results and discussion

Table 1: Please specify in the title the samples on which the Total carotenoids, encapsulation efficiency, and antioxidant activity is reported

Pag 7, line 295: replace with “as a way”

Pag 7, line 299: and throughout the text: Use the full stop to separate the decimal

Pag 7, line 307: “The Arabic gum has structural characteristics that allow lipophilic surfaces to adsorb it”. The sentence is not clear. What is absorbed on the Arabic gum surface? Lipophilic substances? Please rewrite

Figure 1 and 2: The quality of the figures is really bad. Please replace

Pag 8, line 340: rewrite cm -1 as superscript

Pag 8, line 343: if the microencapsulation process had been successful, I would have expected a similar spectrum between the binary mixture (That is the coating material) and the microparticles spectrum. Please better substantiate the results obtained.

Pag 11, line 420: how the PDI was evaluated? In M&M section you speak only of the observation through SEM

Pag 11, line 433: how was the size of the microparticles assessed? On how many particles has it been evaluated?

Author Response

Dear editors and reviewers,

We thank you for the time spent and for all the comments made to the manuscript. We respectfully welcome and appreciate the valuable contributions of the reviewers, with their respective suggestions and requests for correction that certainly qualify and give greater consistency to our text.

Response to Reviewer 3 Comments

Abstract:

Point 1: Pag 1, line 16: Please delete the comma after generates

Response 1: The authors agree with the reviewer. The manuscript has been modified as requested.

Point 2: Pag 1, line 17: Please replace “potencial” with “potential”

Response 2: The manuscript has been modified as requested.

Point 3: Pag 1, line 24: Rewrite as “The antioxidant activity assessed through the ABTS value was ….”

Response 3: The authors amended in the manuscript.

Point 4: Pag 1, line 27-29: Rewrite as “The results showed that the oily extract of the tucumã almond coproduct can be used as a carotenoid-rich source and microencapsulated with possible application for functional foods production

Response 4: The authors agree with the reviewer. The manuscript has been modified as requested.

Point 5: Since most of the microparticle’s properties depend, inter alia, on the coating material selection please justify the choice

Response 5: The authors chose maltodextrin as one of the encapsulating agents due to its characteristics such as low cost, its influence on physicochemical characterization such as moisture content, particle size, hygroscopicity, water activity, solubility, for improving texture, altering flavor, influencing film formation, increasing crystallization, decreasing viscosity, increasing the temperature of vitreous transition of untapped compounds. Initially, the objective was to use only maltodextrin as an encapsulating agent. However, when starting the preparation of the formulations for the drying process it was not possible to obtain an emulsion only with maltodextrin to be submitted to the process. The study sample and the compound of interest, carotenoids, have lipophilic characteristics. In this sense, as the arabic gum has structural characteristics that allow it to be adsorbed on lipophilic surfaces, act as protective colloid, good film-rumouring agent, present low viscosity and Newtonian behavior, we chose to use it also as an encapsulating agent.By joining Arabic gum and maltodextrin it was possible to obtain a homogeneous emulsion possible to be submitted in the drying process and encapsulation by spray dryer.

Point 6: Throughout the text uniform the coproduct or co-product form

Response 6: The authors agree with the reviewer. The manuscript has been modified as requested.

Introduction:

Point 7: Pag 1, line 37: Please replace “use” with “uses”

Response 7: The manuscript has been modified as requested.

Point 8: Pag 1, line 41-43:  Please rewrite the following sentence: However, there are several studies in literature [2,3] in which residues or by- products of plant origin are used as substrates still containing active principles, because that the material which the material that would be thrown away become util and can be reusing in the production chain

Response 8: The authors agree with the reviewer. The manuscript has been modified as requested.

Point 9: Pag 2, line 44: Please close the round bracket

Response 9: The authors agree with the reviewer. The manuscript has been modified as requested.

Point 10: Pag 2, line 47: Seems that the choice of using the verb to be is not appropriate in the sentence Carotenoids are several functions. Please modify.

Response 10: The authors agree with the reviewer.The manuscript has been modified as requested.

Point 11: Pag 2, line 75: Add dash in beta -carotene

Response 11: The authors agree with the reviewer. The manuscript has been modified as requested.

Point 12: Pag 2, line 95: The reference’s style reported is not correct. Please delete the first letter of the authors' name

Response 12: The authors agree with the reviewer. The authors excluded the first letter of the authors’ name in the references as requested.

Point 13: Pag 3, line 98-106: The references style reported in this part is not correct. Please delete the first letter of the authors' name

Response 13: The authors excluded the first letter of the authors’ name in the references as requested.

Point 14: Pag 3, line 110: Replace “by drying spray” with “by spray drying”

Response 14: The authors agree with the reviewer. The manuscript has been modified as requested.

Point 15: Pag 3, line 122:  please uniform the form of Arabic gum in the text.

Response 15: The authors agree with the reviewer. The manuscript has been modified as requested.

Materials and Methods

Point 16: Pag 3, line 133: The title “Sample Obtaining” must be changed to better specify the paragraph content.

Response 16: The authors agree with the reviewer. The manuscript has been modified as requested.

Point 17: Pag 3, line 145: Was the Oily extract obtained after the almonds mechanical press as reported in the previous paragraph? The title of this paragraph is not appropriate. Please refer to the extraction of carotenoids from the co-product

Response 17: Oily extract was not obtained by mechanical pressing. The oily extract was obtained in ultrasonic equipment. Cold pressing was performed to obtain the oil used industrially in the Amazon oil Industry and the resulting product of this process was given for the development of this study, the co-product of the tucumã almonds. The authors agree with the reviewers and made the change in the manuscript.

Point 18: Pag 3, line 139: throughout the text, make the temperatures uniform by adding a space between the number and the symbol (40 °C)

Response 18: The authors agree with the reviewer. The manuscript has been modified as requested.

Point 19: Pag 3, line 140-142: Please rewrite the sentence

Response 19: The authors wrote the sentence and hope that the text has become clearer.

Point 20: Pag 4, line 158: the spelling of solubilized, please correct

Response 20: The authors agree with the reviewer. The manuscript has been modified as requested.

Point 21: Please reverse the paragraphs 2.5-2.6 order. In the paragraphs Encapsulation efficiency, after the explanation of the extraction methods, please report the quantification of the total carotenoids by spectrophotometry

Response 21: The authors agree with the reviewer. The manuscript has been modified as requested.

Point 22: In all the equations reported, please adjust the number. The same equation is marked with two numbers.

Response 22: The authors agree with the reviewer. The manuscript has been modified as requested.

Point 23: Pag 4, line 177: add a space between the number and the symbol (0.4 g)

Response 23: The authors agree with the reviewer. The manuscript has been modified as requested.

Point 24: Pag 4, line 186: is there 30 uL referred to the microparticle’s extract? Please correct

Response 24: The authors made changes to the manuscript.

Point 25: Pag 5, line 192: Please delete the full stop in β-carotene word

Response 25: The manuscript has been modified as requested.

Point 26: Pag 5, line 193-205: Please uniform the ml format in all the paragraph

Response 26: The manuscript has been modified as requested.

Point 27: Pag 5, line 214-222: Please add the software used to obtain the FT-IR spectrum

Response 27: The software was added in the manuscript.

Point 28: Pag 5, line 273: Is the amount express as mg/g (as reported in the table) or mg/100 g (as reported in the text.

Response 28: The amount express as mg/g as reported in the table. The manuscript has been modified.

Results and discussion

Point 29: Table 1: Please specify in the title the samples on which the total carotenoids, encapsulation efficiency, and antioxidant activity is reported

Response 29: The authors agree with the reviewer. The samples were specified in title the Table 1.

Point 30: Pag 7, line 295: replace with “as a way”

Response 30: The authors agree with the reviewer. The manuscript has been modified as requested.

Point 31: Pag 7, line 299: and throughout the text: Use the full stop to separate the decimal

Response 31: The authors agree with the reviewer. The manuscript has been modified as requested.

Point 32: Pag 7, line 307: “The Arabic gum has structural characteristics that allow lipophilic surfaces to adsorb it”. The sentence is not clear. What is absorbed on the Arabic gum surface? Lipophilic substances? Please rewrite

Response 32: The authors agree with the reviewer, the sentence is not clear. The sentence was rewrite. An Arabic gum that is absorbed on lipophilic surfaces.

Point 33: Figure 1 and 2: The quality of the figures is really bad. Please replace

Response 33: The authors agree with the reviewer. The figures were replaced in the manuscript.

Point 34: Pag 8, line 340: rewrite cm -1 as superscript

Response 34: The authors agree with the reviewer. The manuscript has been modified as requested.

Point 35: Pag 8, line 343: if the microencapsulation process had been successful, I would have expected a similar spectrum between the binary mixture (That is the coating material) and the microparticles spectrum. Please better substantiate the results obtained.

Response 35: The authors reported successful in the microencapsulation process because the atomization of the extract is generating microparticle with maltodextrin Arabic gum as a wall and the extract inside the microparticles, this design of the microcapsule is the reason because the spectrum MB is so similar spectrum oil extract. In the binary mixture (maltodextrine and Arabic gum and oily extract) the extract is not microencapsulated, so can clearly see characteristic bands of the oily extract. The authors agree with the reviewer and in this sense, modified the discussion of the FTIR in the manuscript.

Point 36: Pag 11, line 420: how the PDI was evaluated? In M&M section you speak only of the observation through SEM

Response 36: The authors revised the manuscript and decided to remove the result of the PDI.

Point 37: Pag 11, line 433: how was the size of the microparticles assessed? On how many particles has it been evaluated?

Response 37: Particle size was evaluated by scanning electron microscopy (SEM) in triplicate (n=3). The authors added in the methodology of scanning electron microscopy.

Reviewer 4 Report

The manuscript entitled “Green Extraction by Ultrasound, Microencapsulation by Spray Drying and Antioxidant Activity of the Tucuma Coproduct (As- trocaryum vulgare Mart.) Almonds” describes the ultrasound extraction of oil from Tucuma industry coproducts, and its spray drying microencapsulation in a mixture of maltodextrin and acacia gum. The antioxidant activity and total carotenoids content of both the oil extract and the microparticles were determined by UV spectroscopy. The microparticles were further characterized by thermogravimetry, infrared spectroscopy and scanning electron microscopy.

The manuscript is well organized and the results presented are promising.

Regarding the content, I would only suggest that the authors included the name of the chemical ABTS (2,2′-azino-di-(3-ethylbenzthiazoline sulfonic acid)) in the abstract and considered if Table 2, presenting the results just for one sample, is really necessary.

Below is a list of minor typos found and English corrections/suggestions:

Line 37 – “uses”, instead of “use”

Line 40 – “in the literature”, instead of “in literature”

Lines 41 to 43 – the sentence “because that the material which the material that would be thrown away become util and can be reusing in the production chain” needs rewriting. It is unclear.

Lines 45 to 46 – There is a missing parenthesis to close the examples of carotenoids. The meaning of “The coproduct tucumã and fruit are very high sources of carotenoids… to be used for carotenoid extraction e utilization by any potential industry” is also a little unclear and should be rephrased.

Line 47 – “Carotenoids have several functions …” instead of “Carotenoids are several functions”

Line 51 – “are naturally lipophilic, so they are easily extracted” instead of “are naturally lipophilic, so easily extracted”

Line 110 – “by spray drying” instead of “by drying spray”

Line 133 – “2.2. Sample Preparation” instead of “2.2. Sample Obtaining”

Line 146 – “To obtain the oily extract” instead of “When the oily extract was obtained”

Lines 147 to 148 – The name of the university is missing in this sentence “of the Federal University of do and submitted to ultrasound…”

Line 154 – “2.4. Microparticle Preparation (MP)” instead of “2.4. Microparticle Obtaining (MP)”

Line 155 – “The microparticles were obtained” instead of “The microparticle was obtained”

Line 157 – “phase with the oily extract.” instead of “phase with oily extract.”

Line 161 – “The next step was the atomization of the emulsion” instead of “The next step was the atomized of the emulsion”

Lines 216 to 217 – “…by mixing the powder sample with potassium bromide (KBr) and pressing it at high pressure …” instead of “…by mixing the powder sample with potassium bromide (KBr) and pressed at high pressure…”

Line 224 – “The TG curves of the oily extract, binary mixture, and microparticles were obtained” instead of “The TG curve of the oily extract, binary mixture, and microcaparticle was obtained”

Line 270 and 296 – “Table 1 shows…” instead of “The table 1 shows…”

Line 271 – “The differences in total carotenoids content of the samples are statistically significant” instead of “The total values carotenoids of the samples are statistically significant different”

Lines 279 to 280 – “…the carotenoid content found in our samples is promising…” instead of “…the carotenoid content found in our samples are promising…”

Line 286 – “(p <0.01)” instead of “(P <0.01)”

Line 300 – “This value is higher…" instead of ”This value, higher…”

Line 330 – “are shown in” instead of “are shows in”

Line 416 – “A morphological analysis” instead of “The morphological analysis”

Author Response

Dear editors and reviewers,

We respectfully welcome and appreciate the valuable contributions of the reviewers, with their respective suggestions and requests for correction that certainly qualify and give greater consistency to our text.

Comments and Suggestions for Authors

The manuscript entitled “Green Extraction by Ultrasound, Microencapsulation by Spray Drying and Antioxidant Activity of the Tucuma Coproduct (As- trocaryum vulgare Mart.) Almonds” describes the ultrasound extraction of oil from Tucuma industry coproducts, and its spray drying microencapsulation in a mixture of maltodextrin and acacia gum. The antioxidant activity and total carotenoids content of both the oil extract and the microparticles were determined by UV spectroscopy. The microparticles were further characterized by thermogravimetry, infrared spectroscopy and scanning electron microscopy.

The manuscript is well organized and the results presented are promising.

We thank you for the time spent and for all the comments made to the manuscript.

Regarding the content, I would only suggest that the authors included the name of the chemical ABTS (2,2′-azino-di-(3-ethylbenzthiazoline sulfonic acid)) in the abstract and considered if Table 2, presenting the results just for one sample, is really necessary.

The authors agree with the reviewer. The authors included the chemical name of the radical ABTS in the abstract. The authors considered the reviewer's observation and concluded that there is no need for the table with only one sample. Therefore, they decided to remove Table 2 from the manuscript.

Response to Reviewer 4 Comments

Below is a list of minor typos found and English corrections/suggestions:

Point 1: Line 37 – “uses”, instead of “use”

Response 1: The authors agree with the reviewer. The manuscript has been modified as requested.

Point 2: Line 40 – “in the literature”, instead of “in literature”

Response 2: The authors agree with the reviewer. The manuscript has been modified as requested.

Point 3: Lines 41 to 43 – the sentence “because that the material which the material that would be thrown away become util and can be reusing in the production chain” needs rewriting. It is unclear.

Response 3:The authors agree with the reviewer, the sentence is unclear and has been rewritten as requested.

Point 4: Lines 45 to 46 – There is a missing parenthesis to close the examples of carotenoids. The meaning of “The coproduct tucumã and fruit are very high sources of carotenoids… to be used for carotenoid extraction e utilization by any potential industry” is also a little unclear and should be rephrased.

Response 4: The parenthesis was included. The authors agree with the reviewer, the sentence unclear. The text has been rephrased in the manuscript.

Point 5: Line 47 – “Carotenoids have several functions …” instead of “Carotenoids are several functions”

Response 5: The authors agree with the reviewer. The manuscript has been modified as requested

Point 6: Line 51 – “are naturally lipophilic, so they are easily extracted” instead of “are naturally lipophilic, so easily extracted”

Response 6: The authors agree with the reviewer. The manuscript has been modified as requested

Point 7: Line 110 – “by spray drying” instead of “by drying spray”

Response 7: The authors agree with the reviewer. The manuscript has been modified as requested

Point 8: Line 133 – “2.2. Sample Preparation” instead of “2.2. Sample Obtaining”

Response 8: The authors agree with the reviewer. The manuscript has been modified as requested

Point 9: Line 146 – “To obtain the oily extract” instead of “When the oily extract was obtained”

Response 9: The authors agree with the reviewer. The manuscript has been modified as requested.

Point 10: Lines 147 to 148 – The name of the university is missing in this sentence “of the Federal University of do and submitted to ultrasound…”

Response 10: The authors agree with the reviewer. The manuscript has been modified as requested

Point 11: Line 154 – “2.4. Microparticle Preparation (MP)” instead of “2.4. Microparticle Obtaining (MP)”

Response 11: The authors agree with the reviewer. The manuscript has been modified as requested

Point 12:Line 155 – “The microparticles were obtained” instead of “The microparticle was obtained”

Response 12: The authors agree with the reviewer. The manuscript has been modified as requested

Point 13: Line 157 – “phase with the oily extract.” instead of “phase with oily extract.”

Response 13: The authors agree with the reviewer. The manuscript has been modified as requested

Point 14: Line 161 – “The next step was the atomization of the emulsion” instead of “The next step was the atomized of the emulsion”

Response 14: The authors agree with the reviewer. The manuscript has been modified as requested

Point 15: Lines 216 to 217 – “…by mixing the powder sample with potassium bromide (KBr) and pressing it at high pressure …” instead of “…by mixing the powder sample with potassium bromide (KBr) and pressed at high pressure…”

Response 15: The authors agree with the reviewer. The manuscript has been modified as requested

Point 16: Line 224 – “The TG curves of the oily extract, binary mixture, and microparticles were obtained” instead of “The TG curve of the oily extract, binary mixture, and microcaparticle was obtained”

Response 16: The authors agree with the reviewer. The manuscript has been modified as requested

Point 17: Line 270 and 296 – “Table 1 shows…” instead of “The table 1 shows…”

Response 17: The authors agree with the reviewer. The manuscript has been modified as requested

Point 18: Line 271 – “The differences in total carotenoids content of the samples are statistically significant” instead of “The total values carotenoids of the samples are statistically significant different”

Response 18: The authors agree with the reviewer. The manuscript has been modified as requested

Point 19: Lines 279 to 280 – “…the carotenoid content found in our samples is promising…” instead of “…the carotenoid content found in our samples are promising…”

Response 19: The authors agree with the reviewer. The manuscript has been modified as requested

Point 20: Line 286 – “(p <0.01)” instead of “(P <0.01)”

Response 20: The authors agree with the reviewer. The manuscript has been modified as requested

Point 21: Line 300 – “This value is higher…" instead of ”This value, higher…”

Response 21: The authors agree with the reviewer. The manuscript has been modified as requested

Point 22: Line 330 – “are shown in” instead of “are shows in”

Response 22: The authors agree with the reviewer. The manuscript has been modified as requested

Point 23: Line 416 – “A morphological analysis” instead of “The morphological analysis”

Response 23: The authors agree with the reviewer. The manuscript has been modified as requested

Round 2

Reviewer 3 Report

No comment